

# Use of machine learning to identify protective factors for death from COVID-19 in the ICU: a retrospective study

Lander Dos Santos[1], Lincoln Luis Silva[2], Fernando Castilho Pelloso[3], Vinicius Maia[4], Constanza Pujals[1], Deise Helena Borghesan[5], Maria Dalva Carvalho[1], Raíssa Bocchi Pedroso[1] and Sandra Marisa Pelloso[1]

[1] State University of Maringá, Graduate Program in Health Sciences, Maringá, Paraná, Brazil
[2] Department of Emergency Medicine, Duke University School of Medicine, Durham, NC, United States of America
[3] Department of Medicine, Federal University of Paraná, Curitiba, Paraná, Brazil
[4] Unicesumar, Maringá, Paraná, Brazil
[5] Union of Catholic Colleges of Mato Grosso, Cuiabá, Mato Grosso, Brazil

Corresponding author
Lincoln Luis Silva,
lincoln.silva@duke.edu,
lincolnluis07@gmail.com

## ABSTRACT

**Background.** Patients in serious condition due to COVID-19 often require special care in intensive care units (ICUs). This disease has affected over 758 million people and resulted in 6.8 million deaths worldwide. Additionally, the progression of the disease may vary from individual to individual, that is, it is essential to identify the clinical parameters that indicate a good prognosis for the patient. Machine learning (ML) algorithms have been used for analyzing complex medical data and identifying prognostic indicators. However, there is still an urgent need for a model to elucidate the predictors related to patient outcomes. Therefore, this research aimed to verify, through ML, the variables involved in the discharge of patients admitted to the ICU due to COVID-19.

**Methods.** In this study, 126 variables were collected with information on demography, hospital length stay and outcome, chronic diseases and tumors, comorbidities and risk factors, complications and adverse events, health care, and vital indicators of patients admitted to an ICU in southern Brazil. These variables were filtered and then selected by a ML algorithm known as decision trees to identify the optimal set of variables for predicting patient discharge using logistic regression. Finally, a confusion matrix was performed to evaluate the model's performance for the selected variables.

**Results.** Of the 532 patients evaluated, 180 were discharged: female (16.92%), with a central venous catheter (23.68%), with a bladder catheter (26.13%), and with an average of 8.46- and 23.65-days using bladder catheter and submitted to mechanical ventilation, respectively. In addition, the chances of discharge increase by 14% for each additional day in the hospital, by 136% for female patients, 716% when there is no bladder catheter, and 737% when no central venous catheter is used. However, the chances of discharge decrease by 3% for each additional year of age and by 9% for each other day of mechanical ventilation. The performance of the training data presented a balanced accuracy of 0.81, sensitivity of 0.74, specificity of 0.88, and the kappa value was 0.64. The test performance had a balanced accuracy of 0.85, sensitivity 0.75, specificity 0.95, and kappa value of 0.73. The McNemar test found that there were no significant differences in the error rates in the training and test data, suggesting good classification.

This work showed that female, the absence of a central venous catheter and bladder catheter, shorter mechanical ventilation, and bladder catheter duration were associated with a greater chance of hospital discharge. These results may help develop measures that lead to a good prognosis for the patient.

## INTRODUCTION

The COVID-19 pandemic, caused by the SARS-CoV-2 virus, has been one of the most significant global public health crises in recent history. Since its appearance in Wuhan, China, in December 2019, more than 758 million cases and 6.8 million deaths have been reported (*World Health Organization , WHO*). The most severe patients often must be admitted to intensive care units (ICUs) (*Phua et al., 2020*). Nevertheless, accurately discerning the prognostic variables crucial for predicting patient discharge amidst the heterogeneous manifestations of COVID-19 poses a great challenge (*Esakandari et al., 2020*). In recent times, several studies have employed machine learning (ML) algorithms to forecast patient outcomes (*Islam et al., 2022*; *Jamshidi et al., 2021*; *Kar et al., 2021*). Thus, diverse sets of variables yield varying model performances. Utilizing a comprehensive set of variables might offer substantial insights for developing a model with optimal performance. Hence, integrating advanced techniques such as ML into prognostic modeling could provide insights for comprehending this complexity and improving patient care strategies.

ICU patients often have critical conditions associated with high morbidity and mortality rates (*Taylor et al., 2021*), including during the COVID-19 pandemic. According to a multicenter study, the death rate from COVID-19 in patients admitted to the ICU was 41.6% (*Armstrong, Kane & Cook, 2020*). Furthermore, studies carried out in Africa have shown a mortality rate of 31.5% (*African COVID-19 Critical Care Outcomes Study Investigators, 2021*), 24% in Europe (*Wendel Garcia et al., 2020*), and almost 40% in Brazil (*Zeiser et al., 2022*), where the high mortality rates of patients hospitalized in UTI are frequently observed in public hospitals (*Soares Pinheiro et al., 2020*).

The main risk factors of mortality involving critically ill patients admitted to the ICU are extended length of stay, age, sex, heart disease, multiple organ failure, sepsis, severe trauma, and acute respiratory failure (*Auld et al., 2022*; *Otero et al., 2020*; *Soares Pinheiro et al., 2020*). Treatment is often challenging as each patient may respond differently; however, in general, treatment may involve the use of advanced life support such as mechanical ventilation (*Fadel et al., 2020*), corticosteroid therapy (*Phua et al., 2020*), renal replacement (*Hajjar et al., 2021*), and use of antiretrovirals (*Beigel et al., 2020*). In addition, continuous monitoring and rigorous clinical evaluation are essential to ensure hemodynamic stability and prevent complications (*Poor et al., 2020*).

Investigating the variables correlated with the death of patients with COVID-19 in the ICU may be fundamental for understanding the disease and improving treatment

strategies (*Liu et al., 2020*; *Pijls et al., 2021*). Data of patients with COVID-19 has been explored utilizing ML to forecast their prognosis (*Kamel et al., 2023*) because it represents a sophisticated and adaptable approach to classification modeling by analyzing large datasets to unveil significant latent relationships or patterns (*Zakariaee et al., 2023*). Studies indicate that, in predicting clinical outcomes among COVID-19 patients, ML methods demonstrate superior accuracy compared to conventional statistical models (*Afrash et al., 2022*).

Several approaches exist to identify variables related to a negative outcome, with the decision tree being one of these methods (*Elhazmi et al., 2022*). The decision tree is a technique based on ML used in medicine capable of identifying and correlating several variables involved with the patient's outcome to assist in elaborating strategies to improve their prognosis (*Giotta et al., 2022*; *Kingsford & Salzberg, 2008*). This analysis can help clinicians identify risk factors for mortality in patients with COVID-19 in the ICU, allowing early and effective interventions to be implemented to improve clinical outcomes.

Knowing that, in Brazil, the number of critically ill patients and mortality in the ICU has reached alarming levels and that the identification of variables involved with discharge is crucial for successful treatment and reduction of mortality, the objective of this research was to verify through machine learning the variables involved in the outcome of patients admitted to the intensive care unit due to COVID-19. The results can help develop more effective preventive and therapeutic measures and patient care, providing better treatment and reducing the negative impact on public health.

## MATERIALS & METHODS

### Study design

This is an observational study with information from the medical records of patients admitted to the intensive care unit due to COVID-19 in a Municipal Hospital in Paraná between March 2020 and July 2021 to verify the clinical aspects related to patient discharged.

### Settings

This study was divided into four stages: (1) Data collection, (2) filtering of variables to remove those that do not meet the criteria, (3) selection of variables with the best performance to explain the outcome, and forth) identification of variables to estimate the risk for the outcome. The steps developed in this study can be seen in Fig. 1.

### Data collection

A spreadsheet was structured containing four groups of variables for data collection. In the first group are the information related to the sociodemographic characteristics of the patients. As for the second group, information about signs, symptoms, results of imaging, and laboratory tests at the time of hospitalization was gathered. The same variables as the second group were also collected for the third group, but three days after hospitalization. Finally, in the fourth group, there are the same variables as in groups 2 and 3, however, obtained at the time of the patient's outcome, that is, if he was discharged from the hospital or died. Patient information was collected using the Epimed Monitor ICU Database®, a platform where all clinical information of patients admitted to the ICU in Brazil is stored
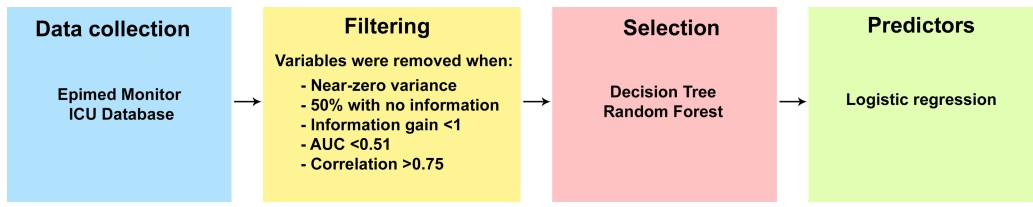

**Figure 1** Steps for identifying variables related to patient outcome.

(*Zampieri et al., 2017*). Data came from electronic case report forms (eCRF) from which information is gathered through integration between electronic records (medical and/or administrative) of the hospital and manual entry of data made by a manager responsible for entering consecutively the information of each patient in the database (*Zampieri et al., 2017*).

## Variables
One hundred twenty-six variables were collected from each patient and grouped into seven groups for a better understanding of the variable's nature, shown in Table 1.

## Participants
The study included all patients over 18 years of age admitted to the ICU of a public municipal hospital, a reference in the care of patients diagnosed with COVID-19, in the state of Paraná, southern region of Brazil.

## Statistical analysis
### Filtering
Prior data processing was initially carried out to eliminate variables with little or no information and redundancies in five steps. First, the variance of the variables was calculated, and those that showed almost zero variance were eliminated from the final set. Second, variables with more than 50% missing values were eliminated, that is, variables without information. Third, the information gain expressed by each variable was identified, and those with values lower than one were discarded. Fourth, the sensitivity and specificity of each variable were analyzed to verify its importance for the model using the receiver operating characteristic (ROC) curve, and those below 0.51 were discarded. Finally, a Pearson correlation analysis was used, and variables with solid correlations, *i.e.,* greater than 0.75, were removed to avoid collinearity. Steps 1, 3, and 4 are described below.

To identify variables with zero variance, those variables with the same value in all lines were checked, and those with almost zero variance when meeting the following requirements: (a) regarding the frequency ratio according to the calculation of the frequency of the highest value prevalent divided by the frequency of the second most prevalent value is greater than 19; and (b) when the percentage of unique values given by the number of unique values divided by the total number of samples (times 100) is less than 10%. Therefore, variables classified as almost zero variance were eliminated.

**Table 1** Variables collected from the Epimed Monitor ICU Database for use in this study. Maringá, Paraná, 2023.

| Classification | Variables |
|---|---|
| Personal | Personal: Age, gender, weight, height (cm); |
| Patient stay and outcome | ICU stay duration, hospital stay duration, unit outcome (treatment/hospitalization outcome); |
| Chronic diseases and tumors | Insuficiência renal crônica não dialítica, insuficiência renal crônica dialítica, cirrose child A ou B, cirrose child C, insuficiência hepática, tumor sólido locorregional, tumor sólido metastático, sítio do tumor, doença hematológica maligna, tipo de doença hematológica maligna, nome da doença hematológica maligna; |
| Comorbidities and risk factors | Immunosuppression, severe COPD, AIDS, systemic arterial hypertension, asthma, uncomplicated diabetes (type 1 or 2), complicated diabetes (type 1 or 2), angina, previous myocardial infarction, arrhythmia, hypothyroidism, hyperthyroidism, peripheral arterial disease, chronic arterial fibrillation, rheumatic disease, sequelae of stroke, stroke without sequelae, dementia, smoking, alcoholism, psychiatric disease, morbid obesity, malnutrition, ischemic heart disease, dyslipidemia, history of pneumonia; |
| Complications and adverse events | Delirium obnubilation stupor or coma, seizure or epilepsy, focal neurological deficit, cardiac rhythm disorders, respiratory failure in the first hour, cardiac arrhythmias in the first hour, cardiopulmonary arrest in the first hour, acute kidney injury in the first hour, asystole in the first hour, pulseless electrical activity in the first hour, atrial fibrillation in the first hour, sustained ventricular tachycardia in the first hour, acute respiratory failure in the first hour, arrhythmias, cardiopulmonary arrest, acute kidney injury, pulseless electrical activity, atrial fibrillation, sustained ventricular tachycardia, non-invasive mechanical ventilation, cardiac rhythm disorders, hypovolemic or hemorrhagic shock, septic shock, anaphylactic or undefined shock, BMI; |
| Healthcare | Failure of non-invasive ventilation, duration of mechanical ventilation, tracheostomy, high-flow mask, duration of hemodialysis, extended hemodialysis in acute kidney injury, hemodialysis, asystole, non-invasive ventilation, vasopressors, hemodialysis in the first hour, mechanical ventilation in the first hour, non-invasive ventilation in the first hour, vasopressors in the first hour, decision on palliative care, chemotherapy, radiotherapy, solid organ transplant, autologous blood transfusion, intestinal transplant, lung transplant, kidney transplant, neurosurgery, central venous catheter, Foley catheter, MAP catheter, intra-aortic balloon, minimally invasive hemodynamic monitoring, red blood cell concentrate transfusion, fresh frozen plasma, thrombolytic agents, NYH classification 2 or 3, use of steroids; |

**Table 1** (*continued*)

| Classification | Variables |
| --- | --- |
| Vital signs | Lowest SBP 1 h, lowest DBP 1 h, lowest MAP 1 h, highest HR 1 h, highest RR 1 h, highest Temperature 1 h, lowest Glasgow Coma Scale 1 h, highest Leukocyte Count 1 h, lowest Platelet Count 1 h, highest Creatinine 1 h, highest Bilirubin 1 h, highest pH 1 h, lowest pH 1 h, highest PaO2 1 h, lowest PaO2 1 h, highest PalCO 21 h, lowest PaCO2 1 h, highest FiO2 1 h, lowest FiO2 1 h, highest PaO2/FiO2 Ratio 1 h, lowest PaO2/FiO2 Ratio 1 h, highest Lactate 1 h, Urea, BUN. |

Algorithms were applied using the method proposed by *Zawadzki & Kosinski (2019)*, which finds important ranks of discrete attributes based on their entropy with a continuous class attribute. Therefore, variables with little or no information gain, that is, values below one, were ignored.

A function called filterVarImp from the caret package in R was used, which applies an algorithm to calculate the two probabilities of dichotomous variables to generate an ROC curve and calculate the area under the curve (AUC) (*Kuhn & Johnson, 2013*). The AUC ranges from 0 to 1 and serves as a measure of variable importance, as an AUC of 0.5 would indicate that a variable cannot discriminate between the two possibilities. An AUC of 1, on the other hand, suggests a variable capable of completely separating the possibilities (*Silveira et al., 2020*). Therefore, those variables with AUC values lower than 0.51 were eliminated.

## Variable selection

For the preparation of decision tree and Random Forest, we segmented the filtered dataset into 80% training and 20% testing sets, to ensure model generalization. This method facilitates an internal validation process: the model is trained on one subset of the data and subsequently tested on another independent subset. This division of the dataset enables an evaluation of the model's capacity to extrapolate predictions to novel instances.

A decision tree is a data analysis technique that uses a set of rules to divide data into smaller, more homogeneous groups. It is a supervised machine learning model used to predict the value of a target variable based on various characteristics. The tree is constructed by dividing the data set into smaller subsets based on the characteristic that best separates the values of the target variable. This division is done through an algorithm where each branch represents a choice or condition that can lead to a specific result (*Venkatasubramaniam et al., 2017*). Then, the subsets are split again until the final leaves of the tree contain enough information to predict the target variable. The goal is to identify patterns and relationships between variables in a data set to make predictions and decisions. When applied to complex problems, the decision tree can help simplify data interpretation and find solutions more efficiently. As a result, the decision tree has been widely used in areas such as medicine (*Hajjej et al., 2022*; *Rokach & Maimon, 2005*).

Subsequently, Random Forest (RF) was employed to identify potential variables associated with patient discharge. The RF classification model underwent training using

10-fold cross-validation on the training dataset. We also used the XGBoost algorithm for predicting patient outcomes and subsequently compared its performance against RF (*Denisko & Hoffman, 2018*; *Montomoli et al., 2021*; *Shanbehzadeh, Nopour & Kazemi-Arpanahi, 2022*). However, the XGBoost algorithm demonstrated inferior performance compared to decision trees, prompting us to maintain the latter as our chosen method for classification. Ultimately, the model's performance was verified using the confusion matrix to evaluate the parameters of sensitivity, specificity, accuracy, kappa agreement test, and McNemar's Test. These variables were then inserted into the logistic regression to determine the chances of the patient progressing to discharge.

All results from the variable selection performed in each of the steps described above are available for further analysis in the supplementary document (https://doi.org/10.6084/m9.figshare.c.6975699).

### Logistics regression

Logistic regression (LR) was employed due to its interpretability, simplicity, and widespread usage in medical research settings, making it feasible to develop prognostic models for COVID-19 patients (*Gutiérrez-Pérez et al., 2022*; *Zapata et al., 2023*). This analysis is an efficient technique that can be used for the classification of two classes. Variables potentially associated with discharge indicated by RF were jointly evaluated using the LR to estimate the odds ratio. Although variables that were highly correlated with each other during data processing were discarded to avoid collinearity problems, variance inflation factor (VIF) was also used in order to detect multicollinearity in the data, and variables with VIF values above ten were removed (*Menard, 2002*). Finally, the quality of the logistic regression fit was verified with the assistance of the half-normal plot and simulated envelope from the "hnp" package in the R software (*Moral, Hinde & Demétrio, 2017*), and the absence of extreme outliers was confirmed through residual analysis. Statistical analyses were conducted using R software version 4.2 (*R Core Team, 2022*) and were considered statistically significant when $p < 0.05$.

### Ethic

This study was approved by the Human Research Ethics Committee of Centro Universitário Ingá according to process no 4.276.900.

### Data availability

Data can be accessed at https://doi.org/10.6084/m9.figshare.c.6975699.

## RESULTS

### Population characteristic

In total, 532 patients were included in the study between March 2020 and July 2021. According to Table 2, most patients were male (57.14%), without severe COPD (93.05%), and with arterial hypertension (70.11%). The majority, 66.92%, of patients did not have complicated diabetes (type 1 or 2), 92.29% did not have complicated diabetes (type 1 or 2), 79.51% were not morbidly obese, and 89.85% did not have hypothyroidism. Regarding

patient care, most did not need non-invasive ventilation in the first hour (68.8%) or vasopressors (62.03%). However, 59.77% required vasopressors, 86.65% a central venous catheter, 90.41% a bladder catheter, and 73.87% an arterial catheter. Only 9.21% needed a transfusion. Regarding relative frequencies, the people who died presented one or more conditions: hypertension, need for vasopressors, central venous catheter, bladder catheter, and/or arterial catheter.

Regarding the continuous variables, in Table 3, it can be seen that the mean and median values for age, highest creatinine at 1 hour, BUN, and duration of mechanical ventilation were lower in the group of patients who were discharged compared to those who died. Conversely, the length of hospital stay had lower mean and median values in the group of deceased patients, possibly because the clinical progression to death is faster than the recovery process in COVID-19 patients admitted to the ICU. Of the 126 variables collected in this study, only 22 remained to classify patients progressing to discharge in order of importance: duration of mechanical ventilation, hospital length stay, age, BUN, highest creatinine 1 h, central venous catheter, bladder catheter, true vasopressors, acute respiratory failure in the first hour, non-invasive ventilation failure, sex, vasopressors in the first hour, uncomplicated diabetes (type 1 or 2), tracheostomy, hypothyroidism, systemic arterial hypertension, non-invasive ventilation in the first hour, morbid obesity, arterial catheter, transfusion, complicated diabetes, and severe COPD. Therefore, these variables were used in logistic regression to estimate the Odds Ratio for discharge. The results of the logistic regression are presented in Table 4.

According to Table 4, the chances of patients being discharged were 136% higher (OR 2.36; 95% CI [1.28–4.44]; $p = 0.007$) in females, 14% higher (OR 1.14; 95% CI [1.10–1.19]; $p < 0.001$) as hospital length stay increases, 737% higher when there was no central venous catheter (OR 8.37; 95% CI [2.41–30.44]; $p < 0.001$), 716% higher when there was no bladder catheter (OR 8.16; 95% CI [2.30–32.14]; $p = 0.002$). However, they decrease by 3% for each additional year of age (OR 1.97; 95% CI [0.95–0.99]; $p = 0.011$), and 9% smaller (OR 0.91; 95% CI [0.86–0.96]; $p = 0.001$) for each additional day of mechanical ventilation.

Table 5 presents the confusion matrix as a performance report of the variables used in RF and logistic regression to verify the variables associated with the patient's discharge. Regarding the training data of RF, the balanced accuracy (proportion of correctly classified cases) could predict 80.5% of the classification, with a sensitivity (true positive rate) of 90.1% and specificity (true negative rate) of 70.9%. The positive predictive value (PPV) expresses the proportion of true positive cases among all positive predictions, which was 84.9%, and the negative predictive value does the same for true negatives (NPV), which was 79.8%. The kappa (a measure of agreement between observed and predicted classifications) was 62.6%, and the McNemar's test (used to compare error proportions between models) was not significant ($p > 0.07$). For the test data, balanced accuracy was 98.2%, sensitivity of 100%, specificity of 96.5%, PPV of 98.4%, NPV of 100%, kappa of 97.4% and McNemar's test was not significative.

Considering the training data, the logistic regression presents a balanced accuracy of 81.7%; that is, the model corrected 81% of the predictions, with a sensitivity of 74.6%
**Table 2** Absolute and relative frequency of clinical variables of patients admitted to the intensive care unit with COVID-19, Maringá, Paraná, 2023.

| Variables | N (%) | | |
|---|---|---|---|
| | **Total** | **Discharge** | **Death** |
| **Sex** | | | |
| Female | 228 (42.86%) | 90 (16.92%) | 138 (25.94%) |
| Male | 304 (57.14%) | 90 (16.92%) | 214 (40.23%) |
| **COPD severe** | | | |
| No | 495 (93.05%) | 171 (32.14%) | 324 (60.9%) |
| Yes | 37 (6.95%) | 9 (1.69%) | 28 (5.26%) |
| **Systemic arterial hypertension** | | | |
| No | 159 (29.89%) | 64 (12.03%) | 95 (17.86%) |
| Yes | 373 (70.11%) | 116 (21.8%) | 257 (48.31%) |
| **Uncomplicated diabetes** | | | |
| No | 356 (66.92%) | 116 (21.8%) | 240 (45.11%) |
| Yes | 176 (33.08%) | 64 (12.03%) | 112 (21.05%) |
| **Complicated diabetes** | | | |
| No | 491 (92.29%) | 169 (31.77%) | 322 (60.53%) |
| Yes | 41 (7.71%) | 11 (2.07%) | 30 (5.64%) |
| **Morbid obesity** | | | |
| No | 423 (79.51%) | 136 (25.56%) | 287 (53.95%) |
| Yes | 109 (20.49%) | 44 (8.27%) | 65 (12.22%) |
| **Hypothyroidism** | | | |
| No | 478 (89.85%) | 164 (30.83%) | 314 (59.02%) |
| Yes | 54 (10.15%) | 16 (3.01%) | 38 (7.14%) |
| **Non-invasive ventilation in the first hour** | | | |
| No | 366 (68.8%) | 121 (22.74%) | 245 (46.05%) |
| Yes | 166 (31.2%) | 59 (11.09%) | 107 (20.11%) |
| **Vasopressors in the first hour** | | | |
| No | 330 (62.03%) | 130 (24.44%) | 200 (37.59%) |
| Yes | 202 (37.97%) | 50 (9.4%) | 152 (28.57%) |
| **Acute respiratory failure in the first hour** | | | |
| No | 329 (61.84%) | 119 (22.37%) | 210 (39.47%) |
| Yes | 203 (38.16%) | 61 (11.47%) | 142 (26.69%) |
| **Vasopressors** | | | |
| No | 214 (40.23%) | 95 (17.86%) | 119 (22.37%) |
| Yes | 318 (59.77%) | 85 (15.98%) | 233 (43.8%) |
| **Failure of non-invasive ventilation** | | | |
| No | 428 (80.45%) | 148 (27.82%) | 280 (52.63%) |
| Yes | 104 (19.55%) | 32 (6.02%) | 72 (13.53%) |
| **Tracheostomy** | | | |
| No | 440 (82.71%) | 136 (25.56%) | 304 (57.14%) |
| Yes | 92 (17.29%) | 44 (8.27%) | 48 (9.02%) |
**Table 2** (*continued*)

| Variables | N (%) | | |
|---|---|---|---|
| | **Total** | **Discharge** | **Death** |
| **Central venous catheter** | | | |
| No | 71 (13.35%) | 54 (10.15%) | 17 (3.2%) |
| Yes | 461 (86.65%) | 126 (23.68%) | 335 (62.97%) |
| **Indwelling bladder catheter** | | | |
| No | 51 (9.59%) | 41 (7.71%) | 10 (1.88%) |
| Yes | 481 (90.41%) | 139 (26.13%) | 342 (64.29%) |
| **PAM catheter** | | | |
| No | 139 (26.13%) | 72 (13.53%) | 67 (12.59%) |
| Yes | 393 (73.87%) | 108 (20.3%) | 285 (53.57%) |
| **Transfusion** | | | |
| No | 483 (90.79%) | 167 (31.39%) | 316 (59.4%) |
| Yes | 49 (9.21%) | 13 (2.44%) | 36 (6.77%) |

**Table 3  Descriptive of continuous variables.**

| Variables | Discharge | Death |
|---|---|---|
| **Age (years)** | | |
| Mean | 57.32 | 63.72 |
| Median | 60 | 65 |
| Standard deviation | 16.71 | 14.01 |
| **Length of hospital stay (days)** | | |
| Mean | 23.65 | 14.08 |
| Median | 17.5 | 12 |
| Standard deviation | 40.24 | 20.99 |
| **Highest creatinine 1 h (mg/dL)** | | |
| Mean | 1.03 | 1.36 |
| Median | 0.7 | 0.8 |
| Standard deviation | 1.43 | 2.07 |
| **BUN (mg/dL)** | | |
| Mean | 28.59 | 37.92 |
| Median | 22.65 | 30.84 |
| Standard deviation | 21.13 | 26.77 |
| **Duration of mechanical ventilation (days)** | | |
| Mean | 8.46 | 9.7 |
| Median | 6 | 8 |
| Standard deviation | 10.2 | 8.12 |

regarding the ability to classify truly positive cases, and the specificity of 88.9% showing the ability to organize negative instances correctly and the kappa value of 64.5%. The test data's balanced accuracy was 85.5%, sensitivity 75.8%, and specificity 95.3%. The kappa value was 73.9%, which indicates a substantial agreement between the predictions and the valid classifications. At the same time, the McNemar test showed no significant differences between the proportions of errors made by the models.

**Table 4** **Multivariate logistic regression to detect variables related to discharge. Maringá, PR, 2023.**

| Variables | Odds ratio | Standard error | 95% IC | p-value |
|---|---|---|---|---|
| (Intercept) | 0.25 | 0.44 | 0.01–7.39 | 0.428 |
| Age | 0.97** | 0.01 | 0.95–0.99 | 0.011 |
| Sex [F] | 2.36*** | 0.75 | 1.28–4.44 | 0.007 |
| Length of hospital stay | 1.14*** | 0.02 | 1.10–1.19 | <0.001 |
| COPD severe [No] | 2.94 | 2.42 | 0.63–15.58 | 0.190 |
| Systemic Arterial Hypertension [No] | 1.19 | 0.42 | 0.60–2.36 | 0.613 |
| Uncomplicated diabetes [No] | 0.91 | 0.30 | 0.48–1.72 | 0.766 |
| Complicated diabetes [No] | 0.70 | 0.42 | 0.22–2.42 | 0.552 |
| Morbid obesity [No] | 0.77 | 0.28 | 0.37–1.60 | 0.469 |
| Hypothyroidism [No] | 2.14 | 1.06 | 0.84–5.96 | 0.124 |
| Non-invasive ventilation in the first hour [No] | 1.38 | 0.54 | 0.64–3.01 | 0.417 |
| Vasopressors in the first hour [No] | 0.95 | 0.48 | 0.35–2.55 | 0.924 |
| Acute respiratory failure in the first hour [No] | 0.49 | 0.20 | 0.22–1.08 | 0.075 |
| Vasopressors [No] | 1.03 | 0.51 | 0.38–2.74 | 0.956 |
| Highest creatinine 1 h | 1.02 | 0.07 | 0.85–1.16 | 0.799 |
| BUN | 1.00 | 0.01 | 0.98–1.01 | 0.677 |
| Non-invasive Ventilation Failure [No] | 0.58 | 0.25 | 0.25–1.36 | 0.212 |
| Duration of mechanical ventilation | 0.91*** | 0.03 | 0.86–0.96 | 0.001 |
| Tracheostomy [No] | 0.47 | 0.21 | 0.19–1.14 | 0.098 |
| Central Venous Catheter [No] | 8.37*** | 5.37 | 2.41–30.44 | 0.001 |
| Indwelling bladder catheter [No] | 8.16*** | 5.40 | 2.30–32.14 | 0.002 |
| PAM catheter [No] | 1.95 | 0.88 | 0.79–4.71 | 0.140 |
| Transfusion [No] | 1.60 | 0.81 | 0.61–4.46 | 0.350 |

**Notes.**
** $p < 0.05$.
*** $p < 0.01$.

# DISCUSSION

Machine learning can identify patterns in the variables involved in the outcome of patients admitted to the intensive care unit due to COVID-19 and propose interventions and decision-making to improve epidemiological data and build care plans.

Investigating and knowing the variables involved in the death and discharge of patients in serious situations are relevant metrics for improving treatment, diagnosis, and quality of care, reducing length of stay and mortality. This is the first study to collect 126 variables to determine associated factors that contributed to the death or discharge of these patients. Among the main factors, the variables, hospital length stay, central venous catheter, and bladder catheter, were related to the discharge of patients admitted to the ICU with COVID-19. In contrast, age and duration of mechanical ventilation were related to greater chances of going to death. The model's performance that detected these variables was satisfactory since the training and test data results are similar.

*Chimbunde et al. (2023)* utilized RF to predict determinants of COVID-19 mortality in South Africa. Surprisingly, their analysis revealed that being female was associated with

**Table 5  Confusion matrix showing the performance of training and test data with the variables used in logistic regression.**

| | Random forest | | | | Logistic regression | | | |
|---|---|---|---|---|---|---|---|---|
| | Train | | Test | | Train | | Test | |
| Outcome | Discharge | Death | Discharge | Death | Discharge | Death | Discharge | Death |
| Discharge | 95 | 24 | 28 | 0 | 100 | 27 | 22 | 3 |
| Death | 39 | 220 | 1 | 64 | 34 | 217 | 7 | 61 |
| **Performance** | | | | | | | | |
| Accuracy | 0.833 | | 0.989 | | 0.838 | | 0.892 | |
| 95% Confidence interval | 0.791–0.869 | | 0.941–0.999 | | 0.797–0.874 | | 0.811–0.947 | |
| Kappa | 0.626 | | 0.974 | | 0.645 | | 0.739 | |
| McNemar's test *p*-value | 0.07 | | 1.000 | | 0.442 | | 0.342 | |
| Precision | 0.849 | | 0.984 | | 0.787 | | 0.880 | |
| Recall | 0.901 | | 1.000 | | 0.746 | | 0.758 | |
| F1 score | 0.874 | | 0.992 | | 0.716 | | 0.814 | |
| Sensitivity | 0.901 | | 1.000 | | 0.746 | | 0.758 | |
| Specificity | 0.709 | | 0.965 | | 0.889 | | 0.953 | |
| Positive predictive value | 0.849 | | 0.984 | | 0.787 | | 0.880 | |
| Negative predictive value | 0.798 | | 1.000 | | 0.864 | | 0.897 | |
| Accuracy balance | 0.805 | | 0.982 | | 0.817 | | 0.855 | |

increased mortality, contrary to findings from our study and others, which suggested that being female was protective (*Elhazmi et al., 2022*; *Kar et al., 2021*). It is noteworthy that their study incorporated different variables than ours. This highlights that the choice of variables can significantly impact the variables identified as associated with the outcome. Moreover, their model exhibited different performance metrics compared to ours, achieving a recall of 76% and a precision of 87%, whereas our model achieved a higher recall of 90.1% and precision of 84.9%, indicating that our model correctly identified more true positives (*Chimbunde et al., 2023*; *Hicks et al., 2022*).

Prolonging hospital length stay was related to patient improvement, and this can be corroborated by *Rees et al. (2020)*, who carried out a systematic review of the length of hospital stay and found that patients who were discharged had a more extended hospital stay than those who died after admission (*Rees et al., 2020*). However, this variable is complex since other studies have detected the opposite. This difference may be related to different criteria used to admit patients to ICUs and the particularities adopted regarding patient care in each location where the studies were conducted (*da Costa Sousa et al., 2022*; *Gupta et al., 2020*).

Non-use of a central venous catheter was another variable correlated with patient discharge. Around 5% to 10% of individuals infected with COVID-19 have clinical conditions that require hospitalization in the ICU and the use of mechanical ventilation (*Pericàs et al., 2020*; *Wu & McGoogan, 2020*). In severe cases, the occurrence rate of acute kidney injury and septic shock is 15% and 20%, respectively (*Ng et al., 2020*). Consequently, administering vasoactive agents or hemodialysis is often necessary, and central venous access is expected (*Chun et al., 2020*). The central venous catheter is a medical device used

in patients who require long-term treatments or large-volume infusions, enabling rapid infusions of volumes and medications that cannot be administered through peripheral vascular access (*Dias et al., 2022*). In this sense, the importance of this instrument in stabilizing the patient and increasing the survival rate is observed. However, not many specific studies have evaluated the use of this instrument in the ICU for patients with COVID-19; on the contrary, some studies have shown a risk of contracting infections and other thrombolytic problems arising from this procedure (*Lugon et al., 2022*).

Another interesting finding in this study was the non-use of the bladder catheter as a predictive variable for patient discharge. Although it is also an instrument capable of evaluating specific renal parameters of patients, it can help doctors make decisions for the patient's clinical improvement. However, it is frequently associated with urinary tract infections in ICUs (*Díaz Pollán et al., 2022*; *Ong et al., 2021*). This underscores the importance of minimizing invasive procedures and implementing strategies to reduce the risk of catheter-associated complications in critically ill patients with COVID-19. Addressing these factors may potentially improve patient outcomes and shorten hospital stays.

Regarding the risks of death, studies have investigated the relationship between age and duration of mechanical ventilation with the death of patients with COVID-19 admitted to the ICU (*Grasselli et al., 2020*; *Richardson et al., 2020*; *Wang et al., 2020*). In general, there is a tendency for older patients to have a higher risk of dying, and patients who require mechanical ventilation for more prolonged periods also have a higher risk of death (*Auld et al., 2020*; *Domecq et al., 2021*). The same was corroborated in this study, in which the average age of death was people over 60 years old.

It was possible through technology to verify the presence of these variables involved in the discharge and deaths of patients admitted to the intensive care unit. This study suggests that paying more attention to these variables is necessary to prevent the condition's progression and increase complications. The use of machine learning in the ICU is evolving but is still limited to diagnostic and prognostic values. *Henry et al. (2015)* developed a machine learning methodology using the MIMIC Clinical Database (Multiparametric Intelligent Monitoring in Intensive Care) –II. They created a model that considers the censoring effects of clinical interventions on the clinical outcomes of patients. This model, called targeted real-time early warning score (TREWSMore), can identify patients at high risk of developing septic shock; it was created in 2015 and is already being applied in some hospitals worldwide (*Henry et al., 2015*; *Niemantsverdriet et al., 2021*). Therefore, it is likely that the use of machine learning tools will become more and more frequent in the future. In this way, this research confirms the findings of previous studies where machine learning demonstrated reliability in determining death and discharge variables.

This study has some limitations. The first is that it is an observational study in which the limits are due to the possibility that some essential variables have not been taken into account or the identification of risk variables. However, more than 126 variables were used here and were carefully selected to create a robust analysis model. The second limitation is that this is a study from a hospital in southern Brazil, which makes it challenging to make generalizations at a global level. However, as technical procedures are standardized

across the national territory, and considering that Brazil has continental dimensions and the diversification of the Brazilian people, it is possible to infer that results can be obtained in other locations within the national territory. Finally, we evaluated the generalizability using the same dataset divided into training and testing sets. This practice is discouraged due to potential demographic biases, as well as biases introduced during data collection, processing, and organization. However, validating the model on additional datasets beyond the original development dataset may not always be feasible due to ethical, technical, or financial constraints associated with sharing clinical data (*Yang, Soltan & Clifton, 2022*).

# CONCLUSIONS

The use of machine learning techniques was possible to identify variables associated with the discharge of patients admitted to the ICU due to COVID-19. The results indicate that factors such as gender, length of hospital stay, presence of central venous catheter, and bladder catheter were significantly related to the likelihood of hospital discharge. Additionally, the study demonstrated that age and duration of mechanical ventilation had a negative impact on the chances of patient discharge. Both the Random Forest and logistic regression models showed satisfactory performance in predicting hospital discharge, with consistent results between training and test data. These findings suggest that the use of ML can be a valuable tool in identifying prognostic factors and supporting clinical decision-making in critically ill COVID-19 patients in the ICU.

## Funding
This study was financed by the Coordenação de Aperfeiçoamento de Pessoal de Nível Superior–Brasil (CAPES)–Finance Code 001. 'There was no additional external funding received for this study. The funders had no role in study design, data collection and analysis, decision to publish, or preparation of the manuscript.

## Grant Disclosures
The following grant information was disclosed by the authors:
The Coordenação de Aperfeiçoamento de Pessoal de Nível Superior–Brasil (CAPES)–Finance Code 001.

## Competing Interests
The authors declare there are no competing interests.

## Author Contributions
- Lander Dos Santos conceived and designed the experiments, authored or reviewed drafts of the article, and approved the final draft.
- Lincoln Luis Silva conceived and designed the experiments, performed the experiments, analyzed the data, authored or reviewed drafts of the article, and approved the final draft.
- Fernando Castilho Pelloso conceived and designed the experiments, prepared figures and/or tables, and approved the final draft.
- Vinicius Maia performed the experiments, prepared figures and/or tables, and approved the final draft.
- Constanza Pujals performed the experiments, analyzed the data, authored or reviewed drafts of the article, and approved the final draft.
- Deise Helena Borghesan analyzed the data, prepared figures and/or tables, and approved the final draft.
- Maria Dalva Carvalho conceived and designed the experiments, authored or reviewed drafts of the article, and approved the final draft.
- Raíssa Bocchi Pedroso conceived and designed the experiments, authored or reviewed drafts of the article, and approved the final draft.
- Sandra Marisa Pelloso conceived and designed the experiments, authored or reviewed drafts of the article, and approved the final draft.

## Human Ethics

The following information was supplied relating to ethical approvals (i.e., approving body and any reference numbers):

The Centro Universitário Ingá granted Ethical approval to carry out the study (Ethical Application Ref: 4.276.900)

## Ethics

The following information was supplied relating to ethical approvals (i.e., approving body and any reference numbers):

The Centro Universitário Ingá granted Ethical approval to carry out the study (Ethical Application Ref: 4.276.900)

## Data Availability

The data is available at figshare: Silva, Lincoln (2023). Use of machine learning to identify protective factors for death from COVID-19 in the ICU. figshare. Collection. Available at https://doi.org/10.6084/m9.figshare.c.6975699.v1.

## Supplemental Information

Supplemental information for this article can be found online at http://dx.doi.org/10.7717/peerj.17428#supplemental-information.

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
