# Peer review of "Use of machine learning to identify protective factors for death from COVID-19 in the ICU: a retrospective study"

_PeerJ, doi:10.7717/peerj.17428_

## Round 0.1 · original submission · Major Revisions

Upon review of your manuscript, several points have been highlighted by the reviewers for your consideration:

-Enhance the introduction and abstract to reflect the focus on machine learning (ML) for COVID-19.
-Justify the choice of logistic regression over other methods and consider comparing different ML algorithms.
-Clearly define the gap addressed by your study in the existing literature.
-Ensure generalizability of results by discussing techniques like cross-validation.
-Consider adding figures like ROC curves or heatmaps for feature importance.
-Elaborate on the study's results and their implications for the field.
-Compare findings with existing literature to highlight contributions.

In addition to the reviewers' comments, please focus specifically on addressing these points in your revisions.

Reviewer 1 ·

Basic reporting

The authors need to add more introduction on the usage of ML for the COVID-19. (See Additional Comments)

Experimental design

The Methods are acceptable. However, why these method (ML model - logistic regression) is used should be discussed. Why not other methods? (See Additional Comments)

Validity of the findings

(See Additional Comments)

Additional comments

1- The title suggests the use of ML while the abstract is far away from it. I suggest that the authors to
complete their abstract with ML results.
2- Same for the introduction. There is no reference on the ML.
3- It might be beneficial to clearly define the gap this study aims to fill in the existing literature.
4- Although various models were mentioned, there is no comparison between them. I suggest that the
authors compare different ML algorithms and report their results.
5- How the authors ensure the generalizability of the results? Did they use cross-validation or
regularization techniques?
6- Please add further figures to enhance the study novelty. For example, I suggest that the authors add
ROC curves for model performance or heat maps for feature importance.
7- The authors need to elaborate on the results of their study and what it adds to the field. For example,
one of the interesting could be if they found any predictive variables that were surprising, and what
might be the clinical implications?
8- Is there anyway that the authors can compare their findings with existing literature and discuss about what their paper add to the field compared to previous ones?

Reviewer 2 ·

Basic reporting

Literature Review: Expand the literature review to provide a more comprehensive background context. This could include more recent studies or reviews on machine learning applications in ICU settings for COVID-19, highlighting the novelty and necessity of the current research.

Data Presentation: Figures and tables should be clearly labeled and described within the text. Ensure that all data presented in figures and tables are directly discussed or referenced in the manuscript to highlight their relevance and importance.

Experimental design

Research Questions and Knowledge Gap: The manuscript should explicitly state the research questions and knowledge gap it aims to address. Clarify how this study contributes uniquely to the existing body of knowledge.
Methodological Detail: Provide more detailed explanations of the machine learning algorithms used, including the rationale behind choosing specific algorithms, any parameter tuning performed, and the validation techniques used to ensure the robustness of the findings.

Validity of the findings

Data Robustness: Discuss the limitations of the data used, such as potential biases, the representativeness of the sample, and how these factors might affect the generalizability of the findings.

Statistical Analysis: Ensure that the statistical methods are adequately described and appropriate for the analysis. For example , "In Random Forest, the Extreme Gradient algorithm was used Boosting (XGBoost) to predict the patient's outcome between discharge or death (28-30)." Random forester (RF) and XGBoost are different ensemble methods for decision trees, i.e. bagging and boosting.

Findings Interpretation: The conclusions drawn should be directly supported by the results presented. Avoid overstating the implications of the findings and discuss any limitations or uncertainties regarding the study's outcomes.

Reviewer 3 ·

Basic reporting

No comments

Experimental design

While the authors suggested a very thorough design for feature cleaning and feature selection, my main concern revolves around the fact that the feature selection, including their AUC analysis and decision tree-based feature selection, was performed on the whole set, not just the train set (that's what I am reading from the steps described sequentially, i.e. the train/test split division is mentioned after the feature selection process is described. Please correct me if I am wrong).

This problem, if indeed present, leads to data leakage and, especially given the large number of variables (126) to select from, may result in overfitting and a lack of generalizability, as the subset of features eventually selected is biased towards the test set.

If the problem is indeed present, I recommend re-deriving the selected feature set on the train set only, fitting the model on the train set, and then reporting the objective test set results. The answer to this question will also determine if minor or major revisions are indeed needed, in my opinion.

I am also not clear what this statement means: 'In Random Forest, the Extreme Gradient algorithm was used Boosting (XGBoost) to predict the patient's outcome between discharge or death (28-30)'. XGBoost and Random Forest are two different algorithms.

Optional Suggestion: For the Random Forest algorithm, you can consider using the out of bag error, rather than cross validation.

Validity of the findings

The model demonstrates a good performance on the test set, however, the feature selection process may need to be reconsidered, as per my comments for the Experimental design section.

---

## Round 0.2 · accepted · Accept

After thorough evaluation by two expert reviewers, the revised manuscript is considered acceptable in its current form. I congratulate the authors for their work.

Reviewer 2 ·

Basic reporting

The author made the proper modifications in the manuscript regarding my questions and comments. I suggest to accept it.

Experimental design

No comment.

Validity of the findings

No comment.

Reviewer 3 ·

Basic reporting

No comments

Experimental design

No comments

Validity of the findings

The issues I previously raised have been thoroughly addressed by the authors, and I appreciate your efforts.